# Concordance rate of radiologists and a commercialized deep-learning solution for chest X-ray: Real-world experience with a multicenter health screening cohort

Eun Young Kim[1◉], Young Jae Kim[2◉], Won-Jun Choi[3], Ji Soo Jeon[2], Moon Young Kim[4,5], Dong Hyun Oh[6,7], Kwang Nam Jin[4,5‡*], Young Jun Cho[6,7‡*]

1 Department of Radiology, Gil Medical Center, Gachon University College of Medicine, Incheon, South Korea, 2 Department of Biomedical Engineering, Gachon University College of Medicine, Incheon, South Korea, 3 Department of Occupational and Environmental Medicine, Gachon University College of Medicine, Incheon, South Korea, 4 Department of Radiology, Boramae Medical Center, Seoul, South Korea, 5 Seoul National University College of Medicine, Seoul, South Korea, 6 Department of Radiology, Konyang University Hospital, Daejeon, Korea, 7 Konyang University School of Medicine, Daejeon, Korea

◉ These authors contributed equally to this work.
‡ KNJ and YJC also contributed equally to this work.
* wlsrhkdska@gmail.com (KNJ); cyj126@kyuh.ac.kr (YJC)

**Data Availability Statement:** All relevant data are within the manuscript and its Supporting Information files.

## Abstract

### Purpose

Lunit INSIGHT CXR (Lunit) is a commercially available deep-learning algorithm-based decision support system for chest radiography (CXR). This retrospective study aimed to evaluate the concordance rate of radiologists and Lunit for thoracic abnormalities in a multicenter health screening cohort.

### Methods and materials

We retrospectively evaluated the radiology reports and Lunit results for CXR at several health screening centers in August 2020. Lunit was adopted as a clinical decision support system (CDSS) in routine clinical practice. Subsequently, radiologists completed their reports after reviewing the Lunit results. The DLA result was provided as a color map with an abnormality score (%) for thoracic lesions when the score was greater than the predefined cutoff value of 15%. Concordance was achieved when (a) the radiology reports were consistent with the DLA results ("accept"), (b) the radiology reports were partially consistent with the DLA results ("edit") or had additional lesions compared with the DLA results ("add"). There was discordance when the DLA results were rejected in the radiology report. In addition, we compared the reading times before and after Lunit was introduced. Finally, we evaluated systemic usability scale questionnaire for radiologists and physicians who had experienced Lunit.

### Results

Among 3,113 participants (1,157 men; mean age, 49 years), thoracic abnormalities were found in 343 (11.0%) based on the CXR radiology reports and 621 (20.1%) based on the

**Funding:** This work was supported by a grant from the Korea Health Industry Development Institute to YJC (Grant number: HI19C0847). The funder had no role in study design, data collection and analysis, decision to publish, or preparation of the manuscript.

**Competing interests:** KNJ has received research grant funding from Lunit Inc. for activities not related to the present article. This does not alter our adherence to PLOS ONE policies on sharing data and materials. Other authors have no potential conflicts of interest to disclose.

Lunit results. The concordance rate was 86.8% (accept: 85.3%, edit: 0.9%, and add: 0.6%), and the discordance rate was 13.2%. Except for 479 cases (7.5%) for whom reading time data were unavailable (n = 5) or unreliable (n = 474), the median reading time increased after the clinical integration of Lunit (median, 19s vs. 14s, $P < 0.001$).

## Conclusion

The real-world multicenter health screening cohort showed a high concordance of the chest X-ray report and the Lunit result under the clinical integration of the deep-learning solution. The reading time slight increased with the Lunit assistance.

## Introduction

The data-intensive nature of medicine makes it one of the most promising fields for the application of artificial intelligence (AI) and machine learning algorithms [1]. Health care centers have become increasingly interested in implementing AI-enabled clinical decision support systems (CDSSs) to improve efficiency and patient outcomes [2]. The system may improve the accuracy and inter-reader variability of physicians in making diagnoses, as well as medical care in resource-constrained environments where healthcare experts are not available. However, there are currently limited examples of successful implementation of AI techniques in clinical practice, and it is not clear how AI tools can be effectively integrated with human decision-making.

Lunit INSIGHT CXR (Lunit) is a commercially available deep-learning algorithm-based CDSS for the automatic detection of thoracic abnormalities on chest X-ray (CXR). Recent studies have reported that AI systems using deep learning techniques can detect various diseases on CXRs, showing performance comparable to that of expert radiologists [3–9]. In previous studies, Lunit showed excellent diagnostic performance, which was similar to that of expert radiologists, and improved the performance of physicians in diagnosing pneumonia, lung cancer, tuberculosis, and multiple abnormal findings [6, 10, 11]. Based on this evidence, Lunit was approved by the Korean Ministry of Food and Drug Safety, and several hospitals have adopted it in routine clinical practice as a decision support system for radiology. However, to the best of our knowledge, no study has evaluated the extent to which radiologists accept the Lunit results in real-world clinical practice. Accordingly, the purpose of this study was to evaluate the concordance of radiology reports and Lunit results for thoracic abnormalities on CXR using a multicenter health screening cohort. In addition, we wanted to compare the reading times before and after the clinical integration of the AI system.

## Materials and methods

This retrospective cohort study was approved by the institutional review boards of three participating institutions (approval number: GBIRB2020-413 for Gil Medical Center, 10-2020-227 for Boramae Medical Center, 2020-10-015-001 for Konyang University Hospital). All the data were de-identified, and the requirement for written informed consent was waived. In the health screening centers of three institutions, Lunit has been adopted in clinical practice since March 2020.

We present the following article based on the Strengthening the Reporting of Observational Studies in Epidemiology (STROBE) reporting checklist (S1 Appendix).

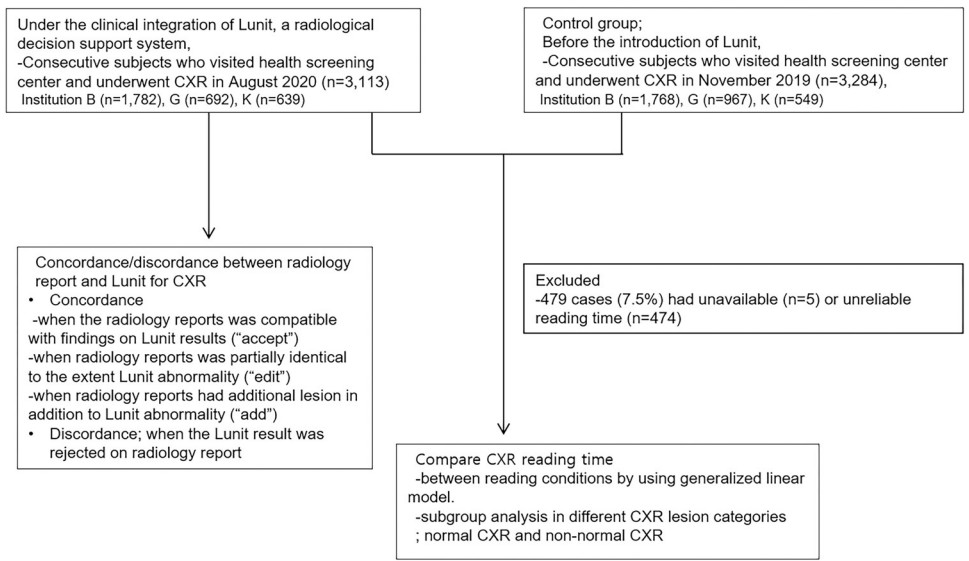

**Fig 1. Flow chart of the study population.** CXR = chest radiography.

## Study population

The data of 3,113 consecutive participants, who visited the health screening center of three institutions and underwent CXR in August 2020, were retrieved from the radiology database and medical records system and retrospectively analyzed. The data of the participants in the control group (n = 3,284), who visited the health screening center the previous year (November 2019) before the clinical integration of the Lunit assistance, were also collected and used to compare the reading times for CXR.

Data on age, sex, and smoking history (pack-years) were retrospectively collected. Based on age and smoking history, the cohort was classified as having a high-risk of lung cancer (aged 55–74 years with ≥30 pack-years of smoking history). Fig 1 shows a flowchart of the study population.

## Radiology report for chest radiographs

The original clinical radiology reports by three board-certified radiologists from three health screening centers (one per institution; C.S.Y., K.R.H., K.S., with 11, 7, and 20 years of experience in radiology, respectively) were retrospectively analyzed. In practice, radiologists evaluate CXR images and review the Lunit results, which are shown as secondary captured images in the reading workstation (picture archiving and communication system PACS, INFINITT Healthcare), to complete the radiology report. Using the original radiology reports, the cases were retrospectively re-categorized by one adjudicator (K.E.Y. with 12 years of experience in thoracic imaging), based on the following semantic descriptions in the radiology reports into normal, inactive lesion, insignificant abnormal lesion, or significant abnormal lesion. Inactive lesions were described as "calcified", "adhesion", "sequelae", "linear", "pleural thickening", "bulla", "s/p pneumonectomy". Insignificant abnormal lesions included "bronchiectasis", "interstitial opacity", "interstitial lung disease", "tiny nodule", and "emphysema". Finally, "focal increased opacity", "nodule", and "consolidation" were allocated to the significant abnormal lesions. A normal CXR (CXR category 0) was categorized as "CXR negative", and the inactive lesions, insignificant abnormal lesions, and significant abnormal lesions (CXR

category 1–3) were categorized as "CXR positive". The descriptions unrelated to lung lesions ("elevation of the diaphragm", "scoliosis", "kyphosis", "bone island", "old rib fracture", "bone cement", "cardiomegaly", "situs inversus", "right-side aortic arch", "prominent pericardial fat pad" and "nipple shadow") were not included as positive findings in the radiology report.

## Lunit for chest radiographs

We used a commercially available deep learning algorithm (Lunit INSIGHT for Chest Radiography Version 2.5.7.4; Lunit, Seoul, South Korea). This version was developed for the detection of three major radiologic findings (nodule/mass, consolidation, and pneumothorax) using a deep convolutional neural network [6]. The raw pixel map of the DICOM images was passed through 34 convolutional layers (ResNet-34 based architecture) that served as feature extractors for projecting the CXR to a good representation space. This was followed by four one-by-one convolution heads that create a color map of each of the four findings. Pixel-wise binary cross-entropy loss and image-level binary cross-entropy loss were used during the training of the model. AI-detected thoracic lesions were marked using a heatmap with an abnormality score (%). The abnormality score indicated the probability (0–100%) that the CXR contained malignant nodule/mass, consolidation, or pneumothorax. Using a predefined cut-off value of 15%, which showed high sensitivity (95%) in internal verification studies [11], lesions with an abnormality score of 15% or more were categorized as "Lunit positive" The Lunit results were integrated with separate images from the original CXR images of the patient in PACS. To complete the radiology report, radiologists reviewed the original CXR image and checked the results of the Lunit integrated as a secondary image.

## Concordance rate for radiology report and Lunit

We determined whether the Lunit results were described in the radiology reports. The cases showed that the CXR negative/Lunit negative cases were designated as "accept", and the CXR positive/Lunit negative or CXR negative/Lunit positive cases were designated as "reject". For the CXR positive/Lunit positive cases, the designations were based on the lesions as follows: accept, when the lesions described on CXR report and Lunit result were in agreement; edit, when the lesions in the radiology reports were in partial agreement with those detected by Lunit; add, the radiology reports had additional lesions compared with those detected by Lunit. When the CXR lesion was different from the Lunit lesion, the case was designated as "reject". The "accept", "edit", and "add" designations represented concordance, and "reject" represented discordance. We evaluated the concordance rate based on the CXR lesion category.

## Reading time before and after the clinical integration of Lunit

The reading time per case was extracted from a PACS log record and calculated as the duration between the opening time and closure time for creating a radiology report. To exclude the cases that remained open for long durations due to unexpected interruptions, we considered more than 120s as an unreliable reading time because readers may have been interrupted and excluded from the analysis.

## System usability scale

Usability refers to ease of use of software technology and the user interface and attributes commonly described include learnability, efficiency, effectiveness, usefulness, accessibility and user satisfaction [12]. System usability scale (SUS) is a tool for measuring both usability and

learnability for practically any kind of system. The SUS scores calculated from individual questionnaires represent the system usability. SUS yields a single number representing a composite measure of the overall usability of the system being studied. SUS is a Likert Scale which includes 10 questions. A total of 24 radiologists and physicians (n = 14 for radiologists, n = 3 for radiology residents, n = 6 for physician) who had any experienced the Lunit were asked to rank each question from 1 to 5 based on their level of agree; 5 means they agree completely, 1 means they disagree strongly. Scoring involves subtracting 1 from all odd items, and subtracting all even numbered item responses from 5, which scales each item from 0 to 4. The total is multiplied by 2.5 to provide a score out of 100, which is interpreted as a percentile ranking and not as a percentage [13, 14]. According to validation studies, the acceptable SUS score is above than industry standards (i.e. above 68) [13, 15].

## Statistical analysis

The descriptive statistics were calculated using SPSS (ver. 20) and are presented as percentages for categorical variables and as means (± standard deviation) or medians (interquartile range) for continuous variables. The continuous variables were compared using the Student t-test or Mann-Whitney U test, and the categorical variables were analyzed using the two-sided Pearson chi-squared test.

For multiple testing, pairwise comparisons and post-hoc analyses were performed, and the $P$-values were corrected using Bonferroni's method. Statistical significance was set at $P < 0.05$.

The concordance rate was defined as the percentage of the cases designated as "accept", "edit", and "add". The reading time and CXR lesion categories were compared before and after the clinical integration of Lunit. The reading times were compared using a generalized linear model with gamma distribution. A subgroup analysis was used for reading time comparisons for the different CXR lesion categories.

## Results

### Baseline characteristics

Table 1 shows the demographic features of the study participants. Compared with the control group, the experimental group was significantly younger (mean age, 49 ± 15 years vs. 52 ± 15

**Table 1. Demographic information.**

| | | After adoption of Lunit (n = 3,113) | Before adoption of Lunit (n = 3,284) | P value |
|---|---|---|---|---|
| Sex, men | | 1,157 (37.2%) | 1,338 (40.7%) | 0.003 |
| Age (years) | | 49±15 | 52±15 | < 0.001 |
| High-risk of lung cancer[†] | | 129 (4.1%) | 160 (4.9%) | 0.161 |
| CXR | normal | 2,770 (89.0%) | 2,998 (91.3%) | 0.017** |
| | Inactive | 226 (7.3%) | 186 (5.7%) | |
| | insignificant abnormal | 27 (0.9%) | 19 (0.6%) | |
| | significant abnormal | 90 (2.9%) | 81 (2.5%) | |
| Further study recommendation | | 37 (1.2%) | 49 (1.5%) | 0.292 |
| Reading time, median (IQR) [††] | | 19s (36s) | 14s (23s) | < 0.001* |

Note: Except where indicated, data are the mean (± SD) or number (%). SD = standard deviation. IQR = interquartile range. Comparisons of means and proportions of the two groups for demographic information were performed using Student's t-test (*Mann-Whitney U test) and chi-squared tests.

[†]High-risk lung cancer patients: age: 55–74 years and a smoking history of 30 pack-years or more.

[††]Except for missing/unreliable reading time information (n = 479, 7.5%).

**at multiple testing, normal CXR was less frequent at Lunit group compared to control (89.0% vs. 91.3% adjusted $P$-value = 0.015).

**Table 2. Concordance according to radiology report and Lunit for chest radiograph (CXR).**

| | CXR positive (11%) | | CXR negative (89%) |
|---|---|---|---|
| Lunit positive (20.1%) | *n = 284 (9.1%) | | Reject (n = 343,11.0%) |
| | *sub-classified as accept (n = 227, 7.3%), edit (n = 28, 0.9%), add (n = 19, 0.6%), and reject (n = 10, 0.3%) | | |
| Lunit negative (79.9%) | Reject (n = 59, 1.9%) | | Accept (n = 2,427, 78.0%) |

*when the lesions described on CXR report and Lunit result were in coincide ("accept"), when the lesions in the radiology reports were in partial agreement with those detected by Lunit ("edit"), when the radiology reports had additional lesions compared with those detected by Lunit ("add"). When the lesion described on CXR report was totally different from the Lunit lesion, the case was designated as "reject".

years, $P < 0.001$) and had more women (62.8% vs. 59.3%, $P = 0.003$). However, the frequency of participants at a high-risk of lung cancer was not significantly different (4.1% vs. 4.9%, $P = 0.161$). The CXR radiology reports showed that a total of 383 (11.0%) participants had abnormalities, including those with inactive (n = 266, 7.3%), insignificant abnormal (n = 27, 0.9%), and significant abnormal (n = 90, 2.9%) lesions and those recommended for further studies (n = 37, 1.2%). Normal CXRs was less frequent after than before the adoption of Lunit (89.0% vs. 91.3% adjusted $P = 0.015$).

## Concordance rate for radiology report and Lunit

Among the participants (n = 3,113), the radiology reports showed that 343 (11.0%) had positive, and Lunit showed that 621 (20.1%) were positive. The concordance rate was 86.8% (accept: 85.3%, edit: 0.9%, and add: 0.6%), and the discordance rate was 13.2% (Table 2) (Fig 2).

The distribution of the discordance cases (n = 412) were as follows: normal (83.3%), inactive lesions (9.7%), insignificant abnormality (1.5%), and significant abnormality (5.6%). The concordance rate was higher for normal (87.6%) than significant abnormal cases (74.4%, adjusted $P = 0.003$) (Table 3) (Fig 3).

## Reading time for the radiology report

Of all the reading times, 479 cases (7.5%) were unavailable (n = 5) or unreliable (n = 474) and were excluded from the analysis. The median reading time increased after the clinical integration of Lunit (19s for AI support vs. 14s for AI unaided readings, $P < 0.001$).

For the generalized linear model, three factors (Lunit support, radiologists, CXR lesion categories) influenced reading time; the average reading time was higher after the clinical integration of Lunit (before vs. after the clinical integration of Lunit, $P < 0.001$) even after adjustment for the radiologists and CXR lesion categories (Table 4). For the subgroup analysis, the average reading time per case increased by 0.2s when the AI support was leveraged for normal CXR. Conversely, the reading time per case decreased by 0.2s with the use of AI support for the non-normal CXR examinations (inactive lesion, insignificant abnormal lesion, and significant abnormal lesions).

## System usability scale

In the SUS questionnaire, the average SUS score was 77.8 (75.7 for radiologists, 81.7 for radiology residents, 80.8 for physician), which was generally considered an acceptable score for system usability (Table 5).

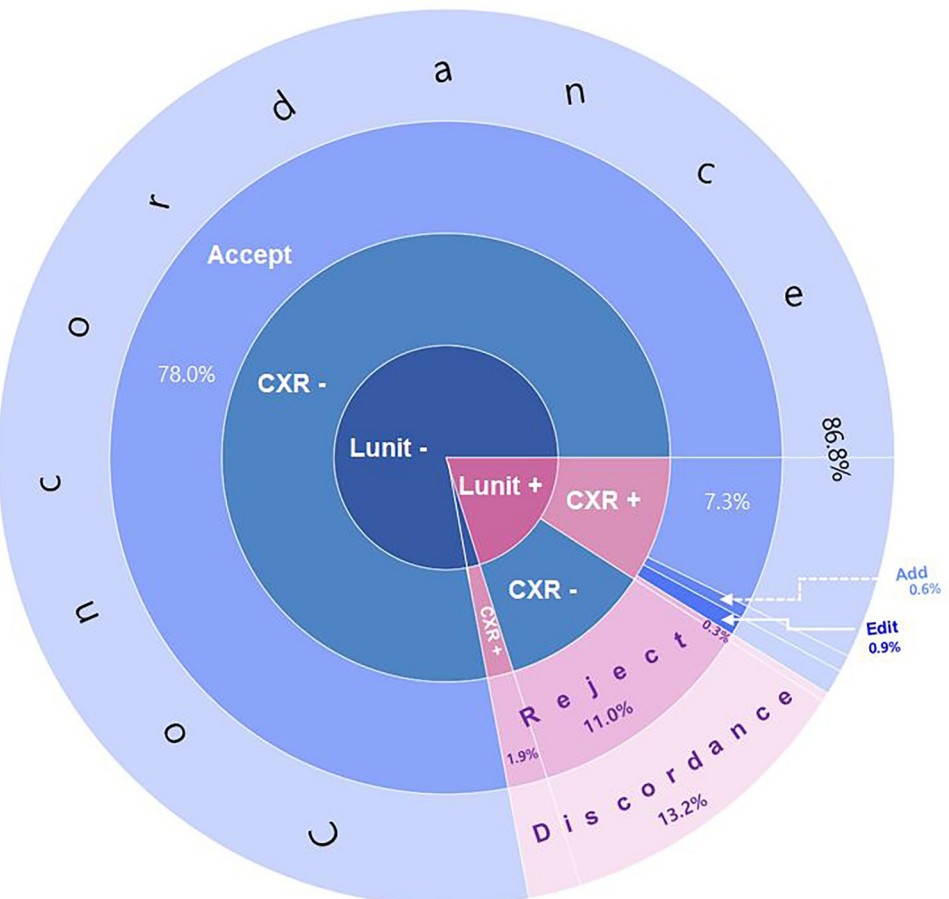

**Fig 2. Sunburst chart for concordance and discordance between Lunit and radiology reports.** Concordance and discordance were based on the agreement of the Lunit results and the radiology report. Concordance was achieved when the radiology reports were consistent with the DLA results ("accept"), the radiology reports were partially consistent with the DLA findings ("edit") or they had additional lesions compared with the DLA findings ("add"). There was discordance when the DLA results were rejected in the radiology report.

## Discussion

Advancements in computer vision and AI have the potential to make significant contributions to health care, particularly in diagnostic specialties such as radiology. However, the perspectives of practicing clinicians and diagnosticians on the integration of AI into medical practice are poorly understood. This study evaluated the concordance rate of radiology reports by radiologists and an AI implementation in real-world clinical practice using a multicenter health

**Table 3. Concordance and discordance of chest radiograph (CXR) report and Lunit result stratified by the CXR lesion category.**

|  | Concordance (n = 2,701) | Discordance (n = 412) | P value |
|---|---|---|---|
| CXR_ normal (n = 2,770; 89.0%) | 2,427 (89.9%) | 343 (83.3%) | < .001* |
| CXR_ inactive (n = 266; 7.3%) | 186 (6.9%) | 40 (9.7%) |  |
| CXR_ insignificant abnormal (n = 27; 0.9%) | 21 (0.8%) | 6 (1.5%) |  |
| CXR_ significant abnormal (n = 90; 2.9%) | 67 (2.5%) | 23 (5.6%) |  |

*at multiple testing, concordance was more frequent for normal CXR than for significantly abnormal CXR (87.6% vs. 74.4% Bonferroni-adjusted P = 0.003).

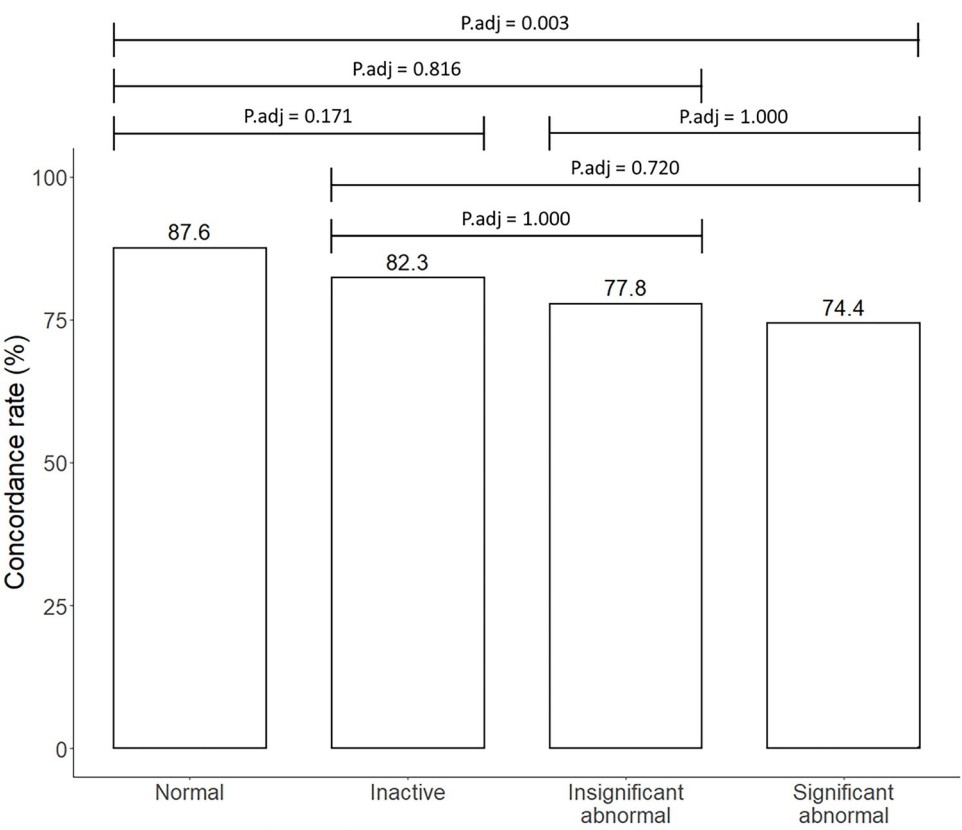

**Fig 3. Concordance rate according to chest radiograph (CXR) lesion categories.** Multiple testing showed that the concordance rate was significantly higher for normal (87.6%) than for significantly abnormal (74.4%, Bonferroni-adjusted $P = 0.003$) cases.

screening cohort. In the health screening cohort, the concordance rate was high (86.8%). Of the discordance case (n = 412, 13.2%), most of the cases are Lunit-positive at normal CXR (83.3%) and followed by Lunit-negative at CXR lesion categories of inactive (9.7%), significant abnormality (5.6%), and insignificant abnormality (1.5%). In addition, the reading time slightly increased with the integration of Lunit assistance in the clinical radiology workflow, compared to the previous year (before the clinical integration of Lunit). Under the Lunit assistance, radiologists should review the Lunit results in addition to their interpretation of the original CXR image to complete the radiology report; the reading time inevitably increased with the use of Lunit. Interestingly, for the CXR-positive cases, the reading time decreased slightly with AI support. This may indicate that AI can facilitate the detection of abnormalities for radiologists. Recently, Lunit was effectively integrated into the PACS workstation system and allows the abnormality score to be directly visible on the PACS worklist screen without opening the resulting images. This enables radiologists to prioritize CXR images as AI normal or abnormal and helps them to read the required images first. This approach is expected to reduce reading time and enhance radiologists' work efficiency.

On the questionnaire, Lunit also reached a reasonable level (average SUS score, 77.8) for the general usability and learnability across the different experience level with CXR and CDSS. SUS has been tried and tested throughout almost 30 years of use, and has proven itself to be a reliable method of evaluating the usability of systems compared to industry standards.

**Table 4. Reading times stratified by the reading condition (before vs. after the clinical integration of Lunit assist), radiologists, and chest radiograph (CXR) lesion categories.**

**For all cases (n = 5,918)**

| Parameters | Estimated | 95% confidence intervals | P-values |
|---|---|---|---|
| Intercept | 3.48 | 3.36, 3.60 | < .001 |
| Lunit aided | 0.19 | 0.15, 0.22 | < .001 |
| Lunit unaided | 0.00 | 0.00 | |
| Radiologist 1 | 1.09 | 1.04, 1.14 | < .001 |
| Radiologist 2 | -0.39 | -0.44, -0.34 | < .001 |
| Radiologist 3 | 0.00 | 0.00 | |
| CXR_normal | -1.13 | -1.24, -1.01 | < .001 |
| CXR_inactive | -0.50 | -0.63, -0.36 | < .001 |
| CXR_insignificant abnormal | -0.12 | -0.35, 0.12 | 0.334 |
| CXR_significant abnormal | 0.00 | 0.00 | |

**For normal CXR (n = 5,362)**

| Parameters | Estimated | 95% confidence intervals | P-values |
|---|---|---|---|
| Intercept | 2.31 | 2.27, 2.36 | < .001 |
| Lunit aided | 0.20 | 0.16, 0.23 | < .001 |
| Lunit unaided | 0.00 | 0.00 | |
| Radiologist 1 | 1.17 | 1.12, 1.22 | < .001 |
| Radiologist 2 | -0.43 | -0.48, -0.37 | < .001 |
| Radiologist 3 | 0.00 | 0.00 | |

**For non-normal CXR (n = 556)**

| Parameters | Estimated | 95% confidence intervals | P-values |
|---|---|---|---|
| Intercept | 3.91 | 3.80, 4.03 | < .001 |
| Lunit aided | -0.16 | -0.25, -0.07 | < .001 |
| Lunit unaided | 0.00 | 0.00 | |
| Radiologist 1 | 0.18 | 0.08, 0.28 | < .001 |
| Radiologist 2 | 0.21 | 0.08, 0.34 | < .001 |
| Radiologist 3 | 0.00 | 0.00 | |
| CXR_inactive | -0.58 | -0.68, -0.48 | < .001 |
| CXR_insignificant abnormal | -0.23 | -0.40, -0.04 | 0.010 |
| CXR_significant abnormal | 0.00 | 0.00 | |

With the high concordance rate and reasonable usability scale indicated, Lunit could be implemented as an assistant systems for CXR interpretation in health screening centers and it might be useful as a training tool and CDSS for unexperienced radiology trainees and physicians.

Based on the evidence of the excellent diagnostic performance of deep learning algorithms, which is comparable to that of an expert radiologist in the health care centers [16] and has power to enhance the physician's performance for the diagnosis of lung cancer, tuberculosis,

**Table 5. System usability scale (SUS) for Lunit.**

| Group | SUS score* |
|---|---|
| All (n = 23) | 77.8 ± 11.9 |
| Radiologists (n = 14) | 75.7 ± 13.8 |
| Radiology residents (n = 3) | 81.7 ± 8.0 |
| Physicians (n = 6) | 80.8 ± 8.5 |

*data are the mean (± standard deviation).

and multiple abnormal findings [6, 10, 11], Lunit was introduced in routine clinical work as an assistance tool for radiology department. Although there is evidence for good diagnostic performance in various clinical settings, the experience of a physician and the attitudes toward the assistance tool can influence how much it embraces the result of the system. Herein, we aimed to evaluate the real-world situation after the adoption of Lunit in health screening centers. We used health care centers from three institutions for this study: two tertiary academic hospitals (located in Incheon and Daejeon in Korea) and one secondary general hospital (in the capital city of Korea, Seoul). In these institutions, Lunit was introduced and has been used as a CDSS in clinical radiology practice since January 2020. The CXR is widely used as a component of periodic health examinations for asymptomatic outpatients or the general population because it has several advantages, including easy accessibility, low cost, and negligible radiation exposure. In Korea, the National Health Service offers free CXR screening biennially to all residents aged 40 years or older [17]. Furthermore, CXR has been widely performed for pre-employment and pre-military service medical screening. The interpretation of CXR is important at health screening setting for the diagnosis of thoracic diseases such as tuberculosis or lung cancer in asymptomatic subjects.

Our study has several limitations. First, we did not evaluate the diagnostic performance of Lunit or radiology report since the primary endpoint of this study was the concordance rate of the radiology report by the radiologist and the Lunit result after its integration into real-world medical practice. For the evaluation of diagnostic performance of Lunit or the radiology reports, the reference standard (ground truth, GT) should be establised based on chest CT or consensus reading by expert radiologists. If we used the chest CT as GT, we could not avoid selection bias, since most of the participants who visited the health clinics did not undergo chest CT examination. To use consensus reading as GT, we needed expert radiologists' time and cost additionally. Since the follow-up data was not sufficient for the participant, we could not use clinical follow-up data as well. However, we wanted to evaluate tremendous number of cases to reflect the real-world situation for the brand-new AI application for CXR, rather than focus to the diagnostic accuracy itself. Second, this study used a specific version of a commercial product with a predefined cut-off value set for high sensitivity. Therefore, careful interpretation is required for the results of the deep learning algorithm for other products or in other clinical settings. Third, the results of our study are limited to one country, so the generalizability to racial differences in other countries is uncertain. Finally, the concordance of AI were evaluated with only three radiologists, which might cause the limited generality. However, it also reflected the real clinical environment that only several radiologists were solely in charge of CXR for health screening centers.

In conclusion, the radiology reports demonstrated high concordance with the results of Lunit, the commercialized AI solution for CXR, in a real-world multicenter health screening cohort. The reading time slight increased after the clinical integration of Lunit support.

## Supporting information

**S1 Checklist. STROBE statement—a checklist of items that should be included in reports of observational studies.**
(DOC)

**S1 Appendix. The concordance rate dataset.**
(XLSX)

## Author Contributions

**Conceptualization:** Eun Young Kim, Young Jae Kim, Kwang Nam Jin, Young Jun Cho.

**Formal analysis:** Eun Young Kim, Young Jae Kim, Won-Jun Choi, Ji Soo Jeon, Kwang Nam Jin.

**Funding acquisition:** Young Jun Cho.

**Methodology:** Eun Young Kim, Young Jae Kim, Kwang Nam Jin, Young Jun Cho.

**Project administration:** Kwang Nam Jin, Young Jun Cho.

**Resources:** Eun Young Kim, Kwang Nam Jin, Young Jun Cho.

**Supervision:** Kwang Nam Jin, Young Jun Cho.

**Validation:** Eun Young Kim.

**Visualization:** Eun Young Kim, Young Jae Kim.

**Writing – original draft:** Eun Young Kim, Young Jae Kim, Kwang Nam Jin.

**Writing – review & editing:** Eun Young Kim, Young Jae Kim, Won-Jun Choi, Ji Soo Jeon, Moon Young Kim, Dong Hyun Oh, Kwang Nam Jin, Young Jun Cho.

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
