## [Decision Letter · Decision Letter 0]

3 Nov 2021

PONE-D-21-19659Concordance rate of radiologists and a commercialized deep-learning solution for chest X-ray: Real-world experience with a multicenter health screening cohortPLOS ONE

Dear Dr. Jin,

Thank you for submitting your manuscript to PLOS ONE. After careful consideration, we feel that it has merit but does not fully meet PLOS ONE’s publication criteria as it currently stands. Therefore, we invite you to submit a revised version of the manuscript that addresses the points raised during the review process.

We look forward to receiving your revised manuscript.

Kind regards,

Alfredo Vellido

Academic Editor

PLOS ONE

Journal Requirements:

2. Thank you for stating the following in the Competing Interests/Financial Disclosure* (delete as necessary) section:

‘KNJ has received research grant funding from Lunit Inc. for activities not related to the present article. This does not alter our adherence to PLOS ONE policies on sharing data and materials. Other authors have no potential conflicts of interest to disclose”

We note that one or more of the authors are employed by a commercial company: name of commercial company.

Reviewers' comments:

Reviewer's Responses to Questions

**Comments to the Author**

1. Is the manuscript technically sound, and do the data support the conclusions?

Reviewer #1: Yes

Reviewer #2: Partly

2. Has the statistical analysis been performed appropriately and rigorously? 

Reviewer #1: Yes

Reviewer #2: Yes

3. Have the authors made all data underlying the findings in their manuscript fully available?

Reviewer #1: Yes

Reviewer #2: No

4. Is the manuscript presented in an intelligible fashion and written in standard English?

Reviewer #1: Yes

Reviewer #2: Yes

5. Review Comments to the Author

Reviewer #1: Summary:

Artificial intelligence (AI) has great potential to fundamentally alter healthcare delivery; however, to data most clinical evaluations commonly concern about artificial elements and often ultimately neglect the human-in-the-loop AI. Success of AI in diagnostic imaging has fueled a growing debate surrounding whether the comprehensive diagnostic interpretive skillsets of radiologist can be replicated by algorithms. This is a nice study timely answer the important question about what’s the role of AI in real-world clinical practice. From this study, we understand that human and AI should be more partners other than competitors. Results of the study revealed high agreement between human and AI in a real-world multicenter health screening cohort.

Limitation:

--There is no follow-up outcome or absence of experts’ consensus reference to evaluate the diagnostic performance of AI. Even the authors argued that they is aimed to evaluate application of AI in a real-world situation rather than focus to the diagnostic accuracy, the accuracy is still first important for a new AI tool in clinical practice. If a new tool is just consistent with experts while did not bring new additional benefit or improvement (such as accuracy, sensitivity or time-saving) to radiologists, we do not know why we use it. So, how is useful of this AI tool to radiologists’ interpretation is yet unknown for us. Please clarify.

--AI were compared with only three radiologists, which is not representative in a real world clinical setting. The interpretation from only three readers may had high variances, potentially impacting the results of study.

Reviewer #2: 1. The manuscript shares 18% similarity with the previously published study which was not cited in the reference:

https://doi.org/10.1371/journal.pone.0246472

2. Proper citation to the previous work is required and clarification must be made whether the data used in this study overlap with the published one.

3. In "Introduction", line 4 - 5 from the bottom, "no study has evaluated the extent to which radiologists accept

the Lunit results in real-world clinical practice". The gap identified here was not addressed in the current study. Different tools must be used for this (e.g., System Usability Scale, UTAUT etc.).

4. According to the PLOS Data policy, it requires authors to make all data underlying the findings described in their manuscript fully available without restriction, with rare exception (please refer to the Data Availability Statement in the manuscript PDF file). The data should be provided as part of the manuscript or its supporting information, or deposited to a public repository. For example, in addition to summary statistics, the data points behind means, medians and variance measures should be available. If there are restrictions on publicly sharing data—e.g. participant privacy or use of data from a third party—those must be specified. No link to the raw data were provided in the manuscript.

5. High concordance result of Lunit is indirectly related to the accuracy of the performance as published in the previous manuscript. It is unclear how this study can contribute more insight to the performance of Lunit. The advantages of the incorporation of Lunit in the existing workflow is also not clearly specified (e.g., can it be used in health screening centers without the assistance of radiologist? Or perhaps as training tools for future or inexperienced radiologist?).

6. PLOS authors have the option to publish the peer review history of their article (what does this mean?). If published, this will include your full peer review and any attached files.

Reviewer #1: **Yes: **Yu-Dong Zhang

Reviewer #2: No

---

## [Author Response · Author response to Decision Letter 0]

19 Nov 2021

19-November-2021

Dear Alfredo Vellido

Thank you very much for the opportunity to revise our original article entitled “Concordance rate of radiologists and a commercialized deep-learning solution for chest X-ray: Real-world experience with a multicenter health screening cohort (PONE-D-21-19659).” After carefully reading the reviewer’ and editor’s comments, we have tried to improve the quality and legibility of the manuscript according to the points raised.

In the revised version, changes are indicated by highlighting. Individual points (E-editor’s comments and R-#1 = point 1 made by reviewer) are indicated in red.

Editor’s comment

PONE-D-21-19659

Concordance rate of radiologists and a commercialized deep-learning solution for chest X-ray: Real-world experience with a multicenter health screening cohort

PLOS ONE

Dear Dr. Jin,

Thank you for submitting your manuscript to PLOS ONE. After careful consideration, we feel that it has merit but does not fully meet PLOS ONE’s publication criteria as it currently stands. Therefore, we invite you to submit a revised version of the manuscript that addresses the points raised during the review process.

We look forward to receiving your revised manuscript.

Kind regards,

Alfredo Vellido

Academic Editor

PLOS ONE

Journal Requirements:

 � Yes, we did.

2. Thank you for stating the following in the Competing Interests/Financial Disclosure* (delete as necessary) section:

“KNJ has received research grant funding from Lunit Inc., outside the present study. This does not alter our adherence to PLOS ONE policies on sharing data and materials. Other authors have no potential conflicts of interest to disclose”

We note that one or more of the authors are employed by a commercial company: name of commercial company.

No authors are employed by a commercial company. 

In cover letter, I add the amended funding statement as follows;

Updated Funding Statement: 

This work was supported by a grant from the Korea Health Industry Development Institute to YJC (Grant number: HI19C0847). The funder had no role in study design, data collection and analysis, decision to publish, or preparation of the manuscript.

In cover letter, I add the COI statement as follows;

KNJ has received research grant funding from Lunit Inc. for activities not related to the

present article. This does not alter our adherence to PLOS ONE policies on sharing data and materials. Other authors have no potential conflicts of interest to disclose.

In cover letter, I add the Data Availability statement as follows;

All relevant data are within the manuscript and its Supporting Information files.

Reviewers' comments:

Reviewer's Responses to Questions 

Comments to the Author

1. Is the manuscript technically sound, and do the data support the conclusions?

Reviewer #1: Yes

Reviewer #2: Partly

2. Has the statistical analysis been performed appropriately and rigorously?

Reviewer #1: Yes

Reviewer #2: Yes

3. Have the authors made all data underlying the findings in their manuscript fully available?

Reviewer #1: Yes

Reviewer #2: No

Thank you for your comment. We added supporting information file.

4. Is the manuscript presented in an intelligible fashion and written in standard English?

Reviewer #1: Yes

Reviewer #2: Yes

5. Review Comments to the Author

Reviewer #1: Summary:

Artificial intelligence (AI) has great potential to fundamentally alter healthcare delivery; however, to data most clinical evaluations commonly concern about artificial elements and often ultimately neglect the human-in-the-loop AI. Success of AI in diagnostic imaging has fueled a growing debate surrounding whether the comprehensive diagnostic interpretive skillsets of radiologist can be replicated by algorithms. This is a nice study timely answer the important question about what’s the role of AI in real-world clinical practice. From this study, we understand that human and AI should be more partners other than competitors. Results of the study revealed high agreement between human and AI in a real-world multicenter health screening cohort.

Thank you for your comments.

Limitation:

1) There is no follow-up outcome or absence of experts’ consensus reference to evaluate the diagnostic performance of AI. Even the authors argued that they is aimed to evaluate application of AI in a real-world situation rather than focus to the diagnostic accuracy, the accuracy is still first important for a new AI tool in clinical practice. If a new tool is just consistent with experts while did not bring new additional benefit or improvement (such as accuracy, sensitivity or time-saving) to radiologists, we do not know why we use it. So, how is useful of this AI tool to radiologists’ interpretation is yet unknown for us. Please clarify.

Thank you for your comments. Many of published data shows comparable diagnostic accuracy of AI. We wanted to show the clinical value for the AI application at real-world setting. In real-world setting, diagnostic accuracy is hard to measure because of 1) hard to get additional consensus reading for too many X-rays 2) if we include cases underwent CT (regarded gold standard test), it cannot avoid selection bias and cannot reflect the real world situation (in real world, only less than 10% underwent CT). 3) FU period was not that enough to show the prognosis of patients.

Nevertheless, we wanted to show how the main users (radiologist) confidently accept the result of AI tool for real-world setting. In addition, we also evaluated the consumption time for CXR reading before and after the adaptation of AI.

Finally, we added System Usability Scale as indicated Reviewer #2, and some discussion.

2) AI were compared with only three radiologists, which is not representative in a real world clinical setting. The interpretation from only three readers may had high variances, potentially impacting the results of study.

Thank you for your comments. We added this point at limitation section.

Reviewer #2: 1. The manuscript shares 18% similarity with the previously published study which was not cited in the reference:

https://doi.org/10.1371/journal.pone.0246472

Thank you for your comments. We added the paper in the reference.

2. Proper citation to the previous work is required and clarification must be made whether the data used in this study overlap with the published one.

Thank you for your comments for our mistake. We added the paper in the reference (as indicated R2-#1). However, the study cohort is totally different and the data used in this study doesn’t overlap at all with data in the previous work. 

3. In "Introduction", line 4 - 5 from the bottom, "no study has evaluated the extent to which radiologists accept the Lunit results in real-world clinical practice". The gap identified here was not addressed in the current study. Different tools must be used for this (e.g., System Usability Scale, UTAUT etc.).

Thank you for your comments. As you indicated, we evaluated how the radiologists confidently accept the result of AI tool for real-world setting, not the general usability of AI. We added the System Usability Scale surveillance for radiologists, radiologist residents and non-radiology physician who were interested in our study. 

4. According to the PLOS Data policy, it requires authors to make all data underlying the findings described in their manuscript fully available without restriction, with rare exception (please refer to the Data Availability Statement in the manuscript PDF file). The data should be provided as part of the manuscript or its supporting information, or deposited to a public repository. For example, in addition to summary statistics, the data points behind means, medians and variance measures should be available. If there are restrictions on publicly sharing data—e.g. participant privacy or use of data from a third party—those must be specified. No link to the raw data were provided in the manuscript.

Thank you for your comment. We added supporting information files.

5. High concordance result of Lunit is indirectly related to the accuracy of the performance as published in the previous manuscript. It is unclear how this study can contribute more insight to the performance of Lunit. The advantages of the incorporation of Lunit in the existing workflow is also not clearly specified (e.g., can it be used in health screening centers without the assistance of radiologist? Or perhaps as training tools for future or inexperienced radiologist?).

Thank you for your comment. We added the description at discussion section.

6. PLOS authors have the option to publish the peer review history of their article (what does this mean?). If published, this will include your full peer review and any attached files.

Do you want your identity to be public for this peer review? For information about this choice, including consent withdrawal, please see our Privacy Policy.

Reviewer #1: Yes: Yu-Dong Zhang

Reviewer #2: No

Thank you again for giving us great honor to resubmit our manuscript to this prestigious journal. We have done our best to respond to all points indicated by the reviewers. We hope you find the revised manuscript acceptable for publication in PLOS ONE.

The authors confirm the manuscript has not been published previously and that it will not be submitted for publication elsewhere and the authors have no conflict of interest to declare.

With best regards,

From Authors

Uploaded:

-Revised and highlighted versions of the revised manuscript (‘Manuscript’ file and

‘Revised Manuscript with Track Changes’ file)

-Rebuttal letter (‘Responses to Reviewers’ file)

---

## [Decision Letter · Decision Letter 1]

4 Jan 2022

PONE-D-21-19659R1Concordance rate of radiologists and a commercialized deep-learning solution for chest X-ray: Real-world experience with a multicenter health screening cohortPLOS ONE

Dear Dr. Jin,

Thank you for submitting your manuscript to PLOS ONE. After careful consideration, we feel that it has merit but does not fully meet PLOS ONE’s publication criteria as it currently stands. Therefore, we invite you to submit a revised version of the manuscript that addresses the points raised during the review process. Please submit your revised manuscript by Feb 18 2022 11:59PM. If you will need more time than this to complete your revisions, please reply to this message or contact the journal office at plosone@plos.org. Please include the following items when submitting your revised manuscript:A rebuttal letter that responds to each point raised by the academic editor and reviewer(s). You should upload this letter as a separate file labeled 'Response to Reviewers'.A marked-up copy of your manuscript that highlights changes made to the original version. You should upload this as a separate file labeled 'Revised Manuscript with Track Changes'.An unmarked version of your revised paper without tracked changes. You should upload this as a separate file labeled 'Manuscript'.If applicable, we recommend that you deposit your laboratory protocols in protocols.io to enhance the reproducibility of your results. Protocols.io assigns your protocol its own identifier (DOI) so that it can be cited independently in the future. For instructions see: https://journals.plos.org/plosone/s/submission-guidelines#loc-laboratory-protocols. Additionally, PLOS ONE offers an option for publishing peer-reviewed Lab Protocol articles, which describe protocols hosted on protocols.io. Read more information on sharing protocols at https://plos.org/protocols?utm_medium=editorial-email&utm_source=authorletters&utm_campaign=protocols.

We look forward to receiving your revised manuscript.

Kind regards,

Alfredo Vellido

Academic Editor

PLOS ONE

Reviewers' comments:

Reviewer's Responses to Questions

**Comments to the Author**

1. If the authors have adequately addressed your comments raised in a previous round of review and you feel that this manuscript is now acceptable for publication, you may indicate that here to bypass the “Comments to the Author” section, enter your conflict of interest statement in the “Confidential to Editor” section, and submit your "Accept" recommendation.

Reviewer #1: (No Response)

Reviewer #2: (No Response)

2. Is the manuscript technically sound, and do the data support the conclusions?

Reviewer #1: Partly

Reviewer #2: Yes

3. Has the statistical analysis been performed appropriately and rigorously? 

Reviewer #1: Yes

Reviewer #2: Yes

4. Have the authors made all data underlying the findings in their manuscript fully available?

Reviewer #1: Yes

Reviewer #2: No

5. Is the manuscript presented in an intelligible fashion and written in standard English?

Reviewer #1: Yes

Reviewer #2: Yes

6. Review Comments to the Author

Reviewer #1: In my initial comments on LIMITATION, I mentioned that only three radiologists were recruited for review, which is not clinically representative. While the response is not satisfactory by just adding a clarification of limitation. The more appropriate anwer is to take a more head to head comparsion betwen different readers, e.g., a team of juniors vs a team of seniors.

Reviewer #2: - Missing data in S2 Appendix, additional data are required for concordance/discordance rate calculation.

- Typographic errors:

1) In "Abstract", misspelling of questionnaire "Finally, we evaluated... questionair...".

2) In "System Usability Scale", misspelling of total, "A toal of 24 radiologists and physicians..." 

- Please check other typo errors throughout the manuscript prior to the final submission.

7. PLOS authors have the option to publish the peer review history of their article (what does this mean?). If published, this will include your full peer review and any attached files.

Reviewer #1: No

Reviewer #2: No

---

## [Author Response · Author response to Decision Letter 1]

18 Jan 2022

18-January-2022

Dear Alfredo Vellido

Happy New Year!

Thank you very much for the opportunity to revise our original article entitled “Concordance rate of radiologists and a commercialized deep-learning solution for chest X-ray: Real-world experience with a multicenter health screening cohort (PONE-D-21-19659).” After carefully reading the reviewer’ and editor’s comments, we have tried to improve the quality and legibility of the manuscript according to the points raised.

In the revised version, changes are indicated by highlighting. Individual points (E-editor’s comments and R-#1 = point 1 made by reviewer) are indicated in red.

PONE-D-21-19659R1

Concordance rate of radiologists and a commercialized deep-learning solution for chest X-ray: Real-world experience with a multicenter health screening cohort

PLOS ONE

Dear Dr. Jin,

Thank you for submitting your manuscript to PLOS ONE. After careful consideration, we feel that it has merit but does not fully meet PLOS ONE’s publication criteria as it currently stands. Therefore, we invite you to submit a revised version of the manuscript that addresses the points raised during the review process.

We look forward to receiving your revised manuscript.

Kind regards,

Alfredo Vellido

Academic Editor

PLOS ONE

Reviewers' comments:

Reviewer's Responses to Questions

Comments to the Author

1. If the authors have adequately addressed your comments raised in a previous round of review and you feel that this manuscript is now acceptable for publication, you may indicate that here to bypass the “Comments to the Author” section, enter your conflict of interest statement in the “Confidential to Editor” section, and submit your "Accept" recommendation.

Reviewer #1: (No Response)

Reviewer #2: (No Response)

2. Is the manuscript technically sound, and do the data support the conclusions?

Reviewer #1: Partly

Reviewer #2: Yes

3. Has the statistical analysis been performed appropriately and rigorously?

Reviewer #1: Yes

Reviewer #2: Yes

4. Have the authors made all data underlying the findings in their manuscript fully available?

Reviewer #1: Yes

Reviewer #2: No

5. Is the manuscript presented in an intelligible fashion and written in standard English?

Reviewer #1: Yes

Reviewer #2: Yes

6. Review Comments to the Author

Reviewer #1: In my initial comments on LIMITATION, I mentioned that only three radiologists were recruited for review, which is not clinically representative. While the response is not satisfactory by just adding a clarification of limitation. The more appropriate anwer is to take a more head to head comparsion betwen different readers, e.g., a team of juniors vs a team of seniors.

Thank you for your comments. We totally agree with you. As you indicated, we evaluate the concordance of AI results with the CXR reading made by only three radiologists, which might not be representative results. However, it also reflected the real clinical environment that only several radiologists were solely in charge of CXR for health screening centers. Actually, there was only one staff radiologist in charge of CXR interpretation in three health care centers.

As a retrospective study, the original clinical radiology reports was made by three board-certified radiologists from three health screening centers (one per institution; C.S.Y., K.R.H., K.S., with 11, 7, and 20 years of experience in radiology, respectively) and we retrospectively analyzed the concordance of AI results and the original radiology reports. 

Reviewer #2: - Missing data in S2 Appendix, additional data are required for concordance/discordance rate calculation.

We checked the data in S2 Appendix. AI result and concordance is not available for control group (group 2, without AI application group) 

- Typographic errors:

1) In "Abstract", misspelling of questionnaire "Finally, we evaluated... questionair...".

2) In "System Usability Scale", misspelling of total, "A toal of 24 radiologists and physicians..."

- Please check other typo errors throughout the manuscript prior to the final submission.

Thank you very much. We amended the typo throughout the manuscript.

7. PLOS authors have the option to publish the peer review history of their article (what does this mean?). If published, this will include your full peer review and any attached files.

Do you want your identity to be public for this peer review? For information about this choice, including consent withdrawal, please see our Privacy Policy.

Reviewer #1: No

Reviewer #2: No

Thank you again for giving us great honor to resubmit our manuscript to this prestigious journal. We have done our best to respond to all points indicated by the reviewers. We hope you find the revised manuscript acceptable for publication in PLOS ONE.

The authors confirm the manuscript has not been published previously and that it will not be submitted for publication elsewhere and the authors have no conflict of interest to declare.

With best regards,

From Authors

Uploaded:

-Revised and highlighted versions of the revised manuscript (‘Manuscript’ file and

‘Revised Manuscript with Track Changes’ file)

-Rebuttal letter (‘Responses to Reviewers’ file)

---

## [Decision Letter · Decision Letter 2]

10 Feb 2022

Concordance rate of radiologists and a commercialized deep-learning solution for chest X-ray: Real-world experience with a multicenter health screening cohort

PONE-D-21-19659R2

Dear Dr. Jin,

We’re pleased to inform you that your manuscript has been judged scientifically suitable for publication and will be formally accepted for publication once it meets all outstanding technical requirements.

Kind regards,

Alfredo Vellido

Academic Editor

PLOS ONE

Additional Editor Comments (optional):

Reviewers' comments:

Reviewer's Responses to Questions

**Comments to the Author**

1. If the authors have adequately addressed your comments raised in a previous round of review and you feel that this manuscript is now acceptable for publication, you may indicate that here to bypass the “Comments to the Author” section, enter your conflict of interest statement in the “Confidential to Editor” section, and submit your "Accept" recommendation.

Reviewer #1: All comments have been addressed

Reviewer #2: All comments have been addressed

2. Is the manuscript technically sound, and do the data support the conclusions?

Reviewer #1: Yes

Reviewer #2: Yes

3. Has the statistical analysis been performed appropriately and rigorously? 

Reviewer #1: Yes

Reviewer #2: Yes

4. Have the authors made all data underlying the findings in their manuscript fully available?

Reviewer #1: Yes

Reviewer #2: Yes

5. Is the manuscript presented in an intelligible fashion and written in standard English?

Reviewer #1: Yes

Reviewer #2: Yes

6. Review Comments to the Author

Reviewer #1: As in R2, the authors can not take a head-to-head comparsion betwen different readers, e.g., a team of juniors vs a team of seniors due to limited readers, I will like to leave this question to editor for judgement.

Reviewer #2: (No Response)

7. PLOS authors have the option to publish the peer review history of their article (what does this mean?). If published, this will include your full peer review and any attached files.

Reviewer #1: No

Reviewer #2: No

---

## [Editor Report · Acceptance letter]

14 Feb 2022

PONE-D-21-19659R2 

Concordance rate of radiologists and a commercialized deep-learning solution for chest X-ray: Real-world experience with a multicenter health screening cohort 

Dear Dr. Jin:

I'm pleased to inform you that your manuscript has been deemed suitable for publication in PLOS ONE. Congratulations! Your manuscript is now with our production department. 

Kind regards, 

on behalf of

Dr. Alfredo Vellido 

Academic Editor

PLOS ONE